# Biochar Application Maintains Photosynthesis of Cabbage by Regulating Stomatal Parameters in Salt-Stressed Soil

Ruixia Chen 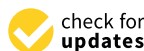, Lijian Zheng, Jinjiang Zhao, Juanjuan Ma * and Xufeng Li

College of Water Resource Science and Engineering, Taiyuan University of Technology, Taiyuan 030024, China
* Correspondence: majuanjuan@tyut.edu.cn

**Abstract:** Salinity is one of the main abiotic stresses, and the stomatal conductance (gs) is an important physiological index in plants that is used to measure their responses to salt stress, as stomatal closure leads to lower yields. However, the factors that affect the gs after biochar application in saline soil remain unclear. To explore the stomatal adaptation mechanisms of facility vegetables under salt stress after the addition of biochar, Chinese cabbage was selected for the pot experiment in this study. The soil and plant $Na^+$ and $K^+$ concentrations, water status, and plant stomatal parameters were measured following treatments with different salt concentrations (25, 50, and 100 mM) and biochar application rates (0, 2, and 4%). The results showed that salt stress induced the physiological closure of the stomata in Chinese cabbage. Compared with the salt-stress treatment without biochar, the biochar application significantly increased the plant gs (20.29–128.79%). Moreover, in the biochar treatment, the open state of the stomata was maintained by sustaining the plant osmotic adjustment, reducing the abscisic acid (ABA) content, and improving the water status. The $Na^+/K^+$ ratio had the most pronounced effect on the stomata (0.702). The actual photochemical efficiency of the photosystem II (ΦPSII) and electron transport rate (ETR) of the Chinese cabbage increased by 0.75–3.41% and 0.65–2.88%, respectively, after the biochar application, which supported the photosynthetic capacity and yield formation. According to the current findings, biochar application can mitigate salt stress and regulate stomatal opening, thereby improving the photosynthesis and the overall yield of Chinese cabbage. Therefore, the application of biochar is a promising method to maintain the productivity of Chinese cabbage under salt stress.

**Keywords:** salt concentration; exogenous additive; stomatal conductance; abscisic acid; osmotic adjustment; leaf turgor

## 1. Introduction

In facility agriculture, inappropriate farming practices are commonly used, which affects the key parameters of the soil properties, such as the electric conductivity (EC) and pH, and which can lead to soil quality degradation [1], which reduces plant yields. In the arid and semiarid areas of China, mulched drip irrigation (MDI) technology has been widely used to improve vegetable production facilities. However, long-term mulched drip irrigation may cause salt accumulation on the soil surface, thereby increasing the risk of soil salinization [2]. Salt stress is one of the main adverse factors that limit plant growth [3]. Salinity inhibits plant development by causing osmotic stress and ion toxicity in plants. The effects of salt stress on plants include physiological and biochemical changes [4]. Some of the effects included may be ion imbalance, increases in the abscisic acid (ABA) content, and the inhibition of water absorption and stomatal opening [5,6]. Leaf stomata are important for the water uptake and carbon assimilation in plants [7]. The reduction in the stomatal conductance (gs) directly leads to a reduction in the final yield [8]. Therefore, maintaining the stomatal opening capacity under salt stress is important for maintaining plant yields.

In recent years, special attention has been paid to the application of nanotechnology and plant biotechnology in agriculture to increase plant yields and alleviate the harms of

environmental stresses [9]. As an exogenous additive, the effect of biochar on the alleviation of salt stress has been demonstrated in tomato, eggplant, and other crops, showing that biochar application can improve the stomatal openings of crops [10,11]. However, we lack information on the factors of the stomatal movement in plants after biochar application. ABA and hydraulic signals are important factors in the regulation of the gs [12,13]. Several studies report that the stomatal closure response to salinity is regulated by the accumulation of ABA in tomatoes, and that most of the decrease in the gs in trees that is caused by water stress can be attributed to changes in the leaf turgor pressure [14,15]. Moreover, the osmotic adjustment ability of plants plays an important role in the maintenance of the cell turgor pressure and stomatal opening [16]. Tardieu et al. [17] showed that osmotic adjustment is the primary controlling factor of the stomatal conductance of plants that grow in saline soil. Therefore, a systematic analysis that combines the plant ABA, turgor pressure, osmotic regulation ability, and gs is helpful for deepening our understanding of the driving factors of the plant stomatal movement induced by biochar application under salt stress.

Turgor pressure is highly correlated with environmental factors. It can be used to assess plant water status more accurately than parameters such as leaf water potential and leaf relative water content [18]. Traditionally, the plant turgor pressure (P) is measured based on the leaf water potential ($\Psi$) and osmotic potential ($\pi$) in a pressure chamber using the formula $P = \Psi - \pi$ [19]. However, this method requires destructive sampling of plants and cannot be used for continuous monitoring [20]. The recently developed non-invasive leaf patch clamp pressure (LPCP) method can precisely measure plant leaf water status in real time [21,22]. The LPCP probe comprises two pads equipped with magnets, one of which contains a pressure-sensor chip. The probe is used to measure the relative changes in leaf turgor pressure by applying a constant pressure using magnets to a small patch of an intact leaf [23]. The application of leaf patch clamp pressure probe technology under salt stress can help us better understand the responses of plants to environmental changes.

Chinese cabbage is a common nutrient-rich vegetable. To date, most studies have focused on the germination and physiological responses of Chinese cabbage under adverse conditions [24,25]. Since relatively little information is available regarding the intrinsic physiological responses of Chinese cabbage to biochar application, our study aimed to investigate whether biochar could alleviate the negative effects of salt stress on gs in Chinese cabbage and to determine the influence factors of biochar on gs in saline soil. We hypothesized that biochar could alleviate the negative effects of salt stress on gs by improving endogenous physiological factors of Chinese cabbage, such as endogenous hormones, osmotic regulation, and water status. This allows it to maintain its photosynthetic capacity. Our results provide a theoretical reference for maintaining the yield of vegetables growing on saline land.

## 2. Materials and Methods

### 2.1. Experimental Materials and Experimental Designs

The experiment was conducted in the greenhouse of Taiyuan University of Technology, Taiyuan, Shanxi Province, from May to June 2022, with an average daily minimum temperature of 15.2 °C, a maximum temperature of 28.3 °C, and an average relative humidity range of 27.3–61.7%. The Chinese cabbage "Shanghai Green" was used as the test material. The seeds were sown in plastic pots ($d_{top}$ = 21 cm, $d_{bottom}$ = 16 cm, and h = 20 cm). Each pot was filled with 4 kg of soil or soil and biochar. The test soil was taken from Xishan, Shanxi Province, and the basic physical and chemical properties of the soil are shown in Table 1. Biochar is made from corn stover. The particle size was 1.5–2 mm, the pH was 9.14, and the mass fractions of carbon, total nitrogen, sulfur, hydrogen and ash were 70.38%, 1.53%, 0.78%, 1.68%, and 31.8%, respectively.

**Table 1.** Physicochemical properties of soil before treatment application.

| Soil Parameters | | Value |
|---|---|---|
| Volume weight (g·cm$^{-3}$) | | 1.35 |
| Field water holding capacity (%) | | 28.8 |
| Total porosity (%) | | 16.78 |
| Organic matter (g·kg$^{-1}$) | | 13.42 |
| pH | | 8.15 |
| Soil machinery composition | Grit 2 to 0.02 mm | 84.05 |
| | Powdered sand grains 0.02 to 0.002 mm | 14.20 |
| | Viscous particles < 0.002 mm | 1.75 |

The experiment was set up considering two factors, namely, saline stress and biochar, with normal soil used as the control (Figure 1A). Based on the tolerance of Chinese cabbage to salt stress [26], three salt concentrations of 25 (S1), 50 (S2), and 100 (S3) mM (Figure 1B) and three biochar concentrations of 0 (B0), 2% (B1), and 4% (B2) (Figure 1C) were used in this study [27–29]. The experiment was conducted with 10 treatments, and 3 replicates were included for each treatment. Saline solution was prepared from NaCl, Na$_2$SO$_4$, NaHCO$_3$, and Na$_2$CO$_3$ at a ratio of 1:9:9:1 [2] (p. 13), [30] (p. 14). The biochar and air-dried soil samples passed through a 2 mm sieve were mixed well in proportion and then potted. In the control group, each pot was filled with 4 kg soil, to which 1200 mL of deionized water was added. In the saline treatment, each pot was filled with 4 kg soil sample, and 1200 mL salt solution was added to prepare the saline soil. For an even distribution of salt, the soil samples were evenly potted four times, and 300 mL solution was poured each time to ensure uniform infiltration of the solution. The pots were then left to stand for 3–4 days. The cabbage seeds were soaked in water and then sown in pots with adequate irrigation. When the seedlings had grown 4–5 leaves, six seedlings of equal size were selected. Based on a maximum daily evapotranspiration of 0.289 mm from a single cabbage plant, irrigation was applied at 2.89 mm intervals every two days to maintain a soil water content of 70–90% of the field water holding capacity.

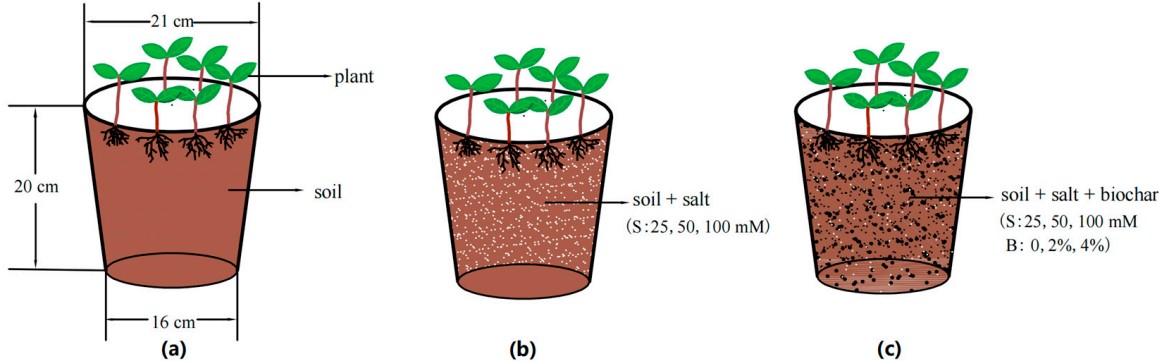

**Figure 1.** Schematic diagram of the pot experiment. (**a**) CK treatment, normal soil without stress and biochar; (**b**) saline stress without biochar addition treatment; (**c**) saline stress with biochar addition treatment. S: salt concentration; B: the proportion of biochar added.

*2.2. Data Recorded*

2.2.1. Determination of Physical and Chemical Properties of the Soil

Soil samples were taken from each treatment after the cabbage harvest. Soil water content was determined using the drying method. Soil samples were dried at 80°C to constant weight. Dry samples were digested with 3:1 (*v/v*) of HNO$_3$ and HClO$_4$. The Na$^+$ and K$^+$ contents in soil were determined using ion chromatography, followed by the method of Liu et al. [31].

### 2.2.2. Determination of Stomatal Indicators in Chinese Cabbage

Stomatal conductance (gs) was measured on fully expanded leaves using a portable photosynthesizer (Li-6400XT) from 9:00–11:00 am. The relative changes of turgor pressure (Pp) of a leaf were measured by the LPCP (ZIM-probe; ZIM Plant Technology GmbH, Hennigs-dorf, Germany). Pp is inversely proportional to leaf turgor pressure, with higher Pp values indicating lower leaf turgor pressure [32].

ZIM-probes were clamped to the non-shaded, expanded leaves of the plant. The surface of the leaf was wiped clean before installation. Although the probes only weigh 5.5 g, cabbage leaves could not support a probe by themselves. Therefore, each probe was supported by a stick impaled in the soil, which allowed the clamped leaf to maintain its original orientation [21]. Data were then recorded every 5 min, with data collection taking place every two days.

Leaf ABA, $Na^+$, and $K^+$ contents were measured after cabbage harvest, and each treatment was replicated three times. Plant leaves were ground for extraction and purification. ABA content was determined using HPLC, according to Huang et al. [33]. Samples of leaves were washed and dried at 80 °C until reaching a constant weight. Dry samples were digested with 3:1 ($v/v$) of $HNO_3$ and $HClO_4$. The $Na^+$ and $K^+$ contents in plant were determined using ion chromatography, followed by the method of Liu et al. [31].

### 2.2.3. Measurement of Photosynthetic Indicators in Chinese Cabbage

The net photosynthetic rate (Pn), transpiration rate (Tr), and intercellular $CO_2$ concentration (Ci) were measured on fully expanded leaves using a portable photosynthesizer (Li-6400XT) from 9:00–11:00 am. Chlorophyll fluorescence was measured on the same leaves used for stomatal conductance measurements using a portable photosynthesizer (Li-6400XT). The fluorescence parameters included the minimum initial fluorescence (Fo), maximum photochemical efficiency of PSII (Fv/Fm), actual photochemical efficiency of PSII (ΦPSII), electron transport rate (ETR), and photochemical quenching coefficient (qP). Fo and Fv/Fm were measured at night after complete darkness, and ΦPSII, ETR, and qP were measured the following morning from 9:00–11:00 am.

### 2.2.4. Yield Measurement

At 30 days after planting, all cabbage plants were weighed in each pot, and the number of plants per pot was recorded. The yield per plant is the ratio of the total weight of each pot to the number of plants.

### 2.3. Statistical Analysis

Statistical analysis was carried out using SPSS 26. Both soil properties and plant parameters were analyzed by two-way ANOVA (analysis of variance) with the factors salt concentration (S), biochar application rate (B), and the interaction of S × B. The significance of differences between mean values was assessed according to the LSD test at the $p < 0.05$ level. Principal component analysis (PCA) and multiple regression analysis were carried out using SPSS 26. The soil volumetric water content (SWC), $Na^+$ and $K^+$ content, and plant $Na^+/K^+$, Pp, ABA, gs, and ΦPSII were selected as the main indicators for PCA. Standardized partial regression coefficients were conducted to test the relationship between $Na^+/K^+$, Pp, ABA, and gs in the biochar treatment.

## 3. Results

### 3.1. $Na^+$ and $K^+$ Concentrations in Soil and Plant

The $Na^+$ concentration in the cabbage significantly increased under salt stress (Table 2, $p < 0.001$), whereas the $K^+$ concentration significantly decreased (Table 2, $p < 0.05$). Under salt stress, the $Na^+$ content increased by 76.66–275.61% in the plants, and the $K^+$ content decreased by 4.16–37.97%, compared with the control treatment. These results led to a significant increase in the $Na^+/K^+$ ratio in the cabbage plants, which reached a maximum value of 2.962 in the S3 treatment (Table 2). The application of biochar reduced

the $Na^+$ content and increased the $K^+$ content in the cabbages. The $Na^+/K^+$ ratio of the cabbage treated with biochar was lower than that of the cabbage treated without biochar. The reductions were 12.01–31.01%, 1.15–18.51%, and 30.26–53.54% in the S1, S2, and S3 treatments, respectively.

**Table 2.** Effects of biochar and salt concentration on soil and cabbage $Na^+$, $K^+$ contents.

| Treatments | Soil | | Plant | | |
| --- | --- | --- | --- | --- | --- |
| | $Na^+$ $(mg \cdot g^{-1})$ | $K^+$ $(mg \cdot g^{-1})$ | $Na^+$ $(mg \cdot g^{-1})$ | $K^+$ $(mg \cdot g^{-1})$ | $Na^+/K^+$ |
| CK | 11.68 ± 1.73 bc | 22.52 ± 3.17 a | 17.20 ± 3.44 c | 38.13 ± 9.60 a | 0.47 ± 0.03 d |
| S1 B0 | 13.43 ± 1.49 abc | 23.31 ± 2.65 a | 30.39 ± 7.14 bc | 35.95 ± 0.54 ab | 0.85 ± 0.21 cd |
| S1 B1 | 13.21 ± 1.00 bc | 23.44 ± 3.57 a | 30.91 ± 7.77 bc | 41.37 ± 7.28 a | 0.75 ± 0.13 d |
| S1 B2 | 14.35 ± 0.40 abc | 24.52 ± 0.33 a | 25.09 ± 3.13 c | 43.74 ± 4.17 a | 0.59 ± 0.11 d |
| S2 B0 | 13.21 ± 0.75 abc | 23.66 ± 2.73 a | 62.57 ± 0.05 a | 36.54 ± 0.76 ab | 1.71 ± 0.04 bc |
| S2 B1 | 11.70 ± 1.11 bc | 23.95 ± 3.38 a | 62.51 ± 8.66 a | 38.18 ± 4.65 a | 1.73 ± 0.41 bc |
| S2 B2 | 16.26 ± 1.02 a | 24.29 ± 0.97 a | 61.26 ± 5.74 a | 43.63 ± 2.13 a | 1.41 ± 0.14 bcd |
| S3 B0 | 13.53 ± 0.15 abc | 23.39 ± 4.09 a | 64.61 ± 3.67 a | 23.65 ± 4.79 b | 2.96 ± 0.58 a |
| S3 B1 | 10.75 ± 1.30 c | 23.67 ± 0.75 a | 62.35 ± 5.55 a | 31.57 ± 1.11 ab | 2.05 ± 0.24 ab |
| S3 B2 | 15.13 ± 2.54 ab | 23.71 ± 3.92 a | 47.64 ± 8.92 ab | 33.10 ± 3.68 ab | 1.42 ± 0.15 bcd |
| *p* values | | | | | |
| S | ns | ns | 0.00 | 0.03 | 0.00 |
| B | 0.01 | ns | ns | ns | 0.04 |
| S × B | ns | ns | ns | ns | ns |

Means within each line followed by the same letter are not significantly different ($p < 0.05$). ±values indicate standard error of the means (SE). S: salt concentration; B: biochar application; S × B: their interactive effect.

The biochar addition had a significant effect on the soil $Na^+$ content (Table 2, $p < 0.05$). We did not observe any significant effects of the salt concentration, biochar addition, or their interaction on the soil $K^+$ content (Table 2, $p > 0.05$). The $Na^+$ and $K^+$ contents in the soil generally increased under salt stress. However, there were no significant differences among the three salt concentrations (Table 2). The effect of the biochar application on the soil ion content varied. The soil $Na^+$ content showed a decreasing trend in the B2 treatment, and particularly in the S2 and S3 treatments, with decreases of 11.46% and 20.55%, respectively. The $K^+$ contents were increased slightly. However, the $Na^+$ and $K^+$ contents were increased obviously in the B4 treatment under all salt concentrations, with the $Na^+$ and $K^+$ contents increasing by 6.85–23.11% and 1.35–5.19%, respectively, compared with the treatment with no biochar application.

### 3.2. Soil Water Content and Leaf Pp Value

Figure 2 shows that the Pp value of the cabbage remained between 49.4 and 54.7 kPa for the control treatment without stress. The overall fluctuation range of the diurnal variation was relatively small. Salt stress significantly increased the Pp values. With the aggravation of salt stress, the Pp values gradually increased, and they reached 70.5–73.5 kPa for the S3B0 treatment (Figure 2a). The application of biochar improved the water status of the cabbage under all levels of salt stress, and the effect was most pronounced under severe salt stress (Figure 2d). Although the Pp value of S2B1 was higher than that of the control without stress, it was lower than that of S2B0 (Figure 2c). The Pp values were reduced by 18.1–37.8%, 6.9–22.9%, and 32.7–42.1% in the S1, S2, and S3 treatments, respectively. B2 was more effective than B1 in all the cases (Figure 2b–d).

The soil water content was measured after the cabbage harvest. The results showed that the salt concentration, biochar addition, and their interaction had no significant effects on the soil volumetric water content (Figure 3a, $p > 0.05$). The addition of salt solution and biochar reduced the soil volumetric water content; however, there was no significant difference between the treatments.

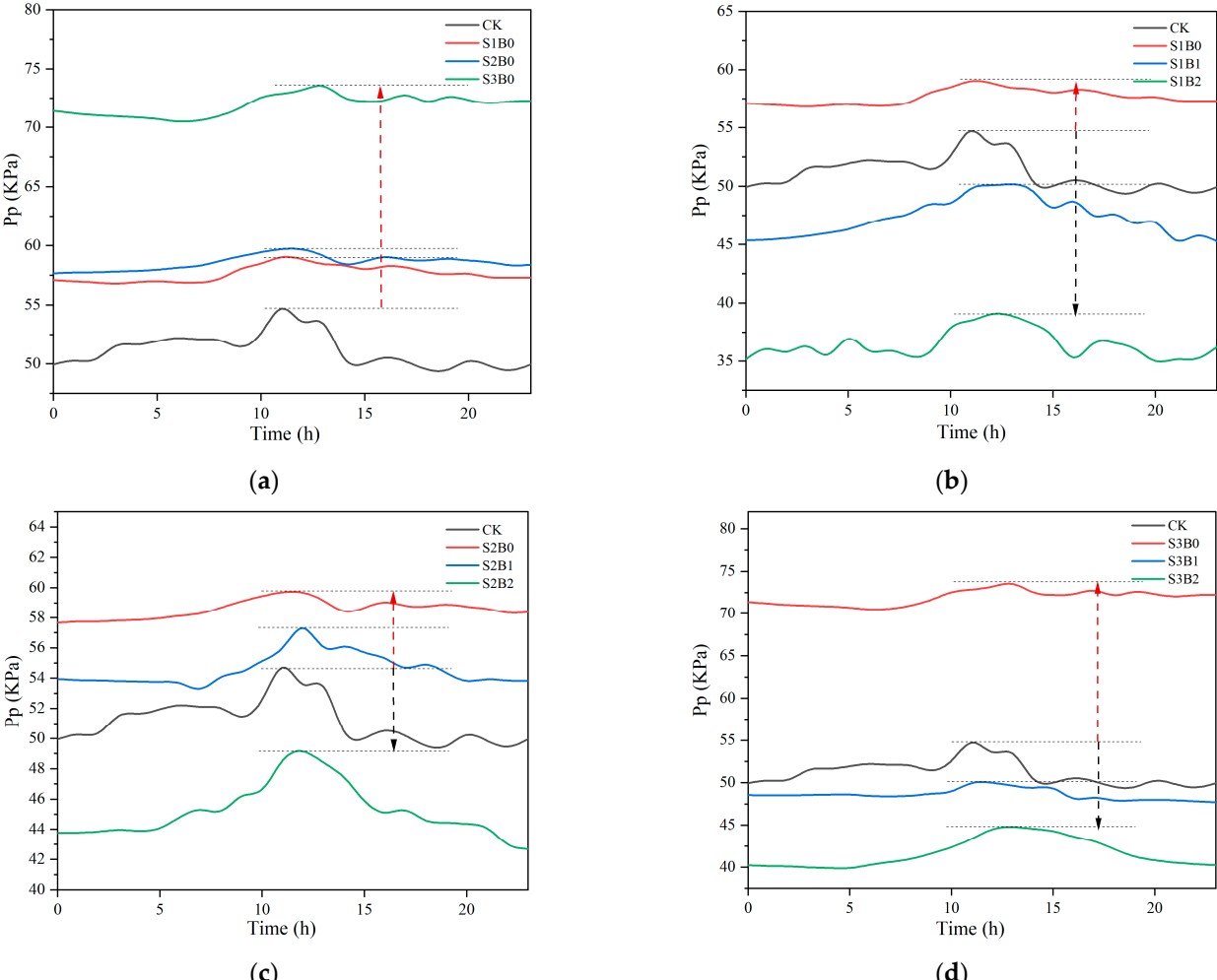

**Figure 2.** The diurnal variation of Pp for (**a**) salt stress without biochar treatments; (**b**) 25 mM salt concentration and biochar addition treatments; (**c**) 50 mM salt concentration and biochar addition treatments; and (**d**) 100 mM salt concentration and biochar addition treatments. The red arrow indicates the change trend of Pp value under salt stress, and the black arrow indicates the change trend of Pp value under the condition of applying biochar.

### 3.3. Leaf ABA Concentrations

The ABA content of the cabbage leaves was significantly affected by the salt stress and biochar addition (Figure 3b, $p < 0.05$). However, their interaction did not have a significant effect on the ABA content ($p > 0.05$). The ABA content gradually increased with the increasing salt concentration, and it significantly increased by 31.27–48.29% compared with the control treatment without stress. The biochar application reduced the ABA content. The more severe the salt stress, the more pronounced the reduction in the ABA caused by the biochar addition (Figure 3b). The addition of different amounts of biochar reduced the ABA contents by 13.87–20.89%, 14.52–17.23%, and 24.29–36.03% in the S1, S2, and S3 treatments, respectively.

### 3.4. Leaf Gas Exchange

Furthermore, the gs was positively correlated with the Pn ($R^2 = 0.74$, $p < 0.001$) (Figure 4b). Decreases in the gs negatively affected the photosynthetic capacities of the plants. We observed that the salt concentration and biochar addition significantly affected the cabbage gas exchange (Figure 4a, $p < 0.05$). Salt stress inhibited the stomatal opening and photosynthesis in the cabbage, and this inhibition increased with the increasing salt-stress

levels. The gs, Pn, Ci, and Tr were reduced by 15.53–59.95%, 32.37–63.59%, 22.08–41.32%, and 7.09–55.49%, respectively, compared with the control treatment without stress. The application of biochar substantially improved the stomatal opening in the cabbage, and it increased the gs by 20.29–128.79% compared with the treatment without biochar. The Pn, Ci, and Tr were increased by 13.70–97.26%, 8.20–36.22%, and 8.20–36.22%, respectively, after the biochar addition.

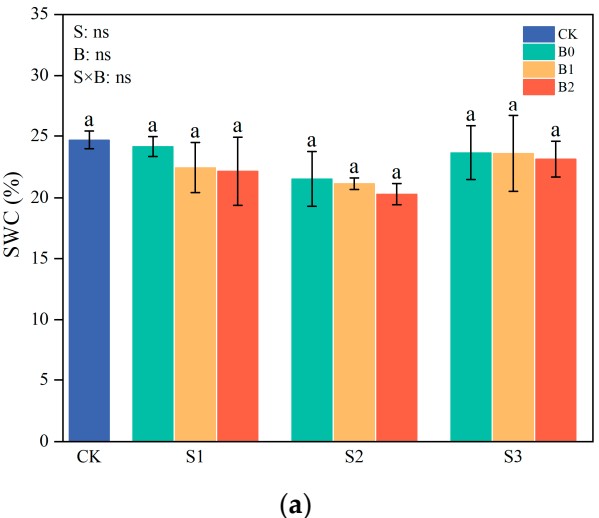

(a)

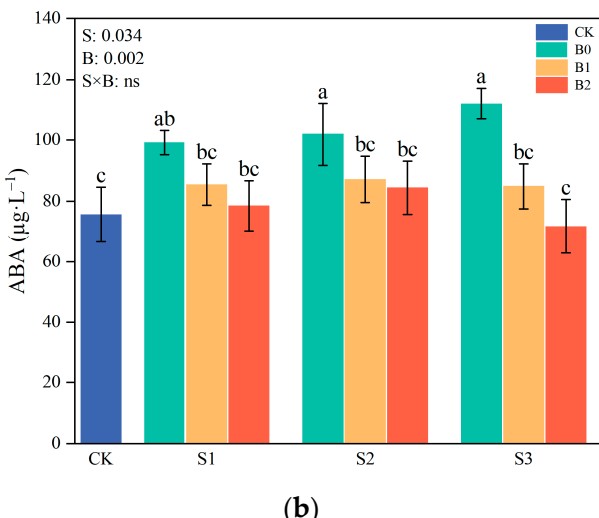

(b)

**Figure 3.** Effect of salt stress and biochar application on the contents of (**a**) soil volumetric water (SWC) and (**b**) abscisic acid (ABA). Bars followed by the same letter are not significantly different ($p < 0.05$). Error bars indicate standard error of the means (SE). S: salt concentration; B: biochar application; S × B: their interactive effect. The upper left digit represents the *p* value of the analysis of variance.

### 3.5. Chlorophyll Fluorescence

The salt concentration had a significant effect on the Fv/Fm, ΦPSII, and ETR ($p < 0.05$). The biochar application significantly affected the Fo, Fv/Fm, ΦPSII, and ETR ($p < 0.05$). There was a significant interaction effect between the salt concentration and biochar application in terms of the ΦPSII and ETR ($p < 0.01$); however, there was no significant interaction effect on the Fo, Fv/Fm, or qP ($p > 0.05$).

Salt stress had a negative effect on the chlorophyll fluorescence parameters in the cabbage (Figure 5a). With the aggravation of the stress levels, the Fo gradually increased, and the Fv/Fm, ΦPSII, ETR, and qP decreased. All the parameters reached minimum values for the S3B0 treatment (except the Fo). From S1 to S3, the Fv/Fm, ΦPSII, and ETR decreased by 0.37–2.35%, 1.35–4.48%, and 2.50–5.06%, respectively, compared with the control treatment. The application of biochar under salt stress improved these parameters (Figure 5b–d). The plants with biochar application had 7.00–11.75% lower Fo values than the plants without the biochar application. The other fluorescence parameters showed opposite trends. The Fv/Fm, ΦPSII, and ETR significantly increased by 1.48–2.26%, 2.91–3.41%, and 2.49–2.88%, respectively, due to the biochar application in the S3 treatment (Figure 5d). The biochar also had a positive effect on the qP at all salt concentrations; however, the differences were not statistically significant.

### 3.6. Plant Yield

The cabbage yield was significantly reduced by 17.36–37.94% under salt stress (Figure 6, $p < 0.05$). This adverse effect was alleviated by the biochar application (Figure 6). The biochar treatments increased the yields by 7.52–22.88%, 13.70–25.90%, and 19.00–34.77% in the S1, S2, and S3 treatments, respectively, showing the overall better effect of the B2 treatment. Furthermore, the interaction between the salt concentration and biochar addition had no significant effect on the yields of individual plants ($p > 0.05$).

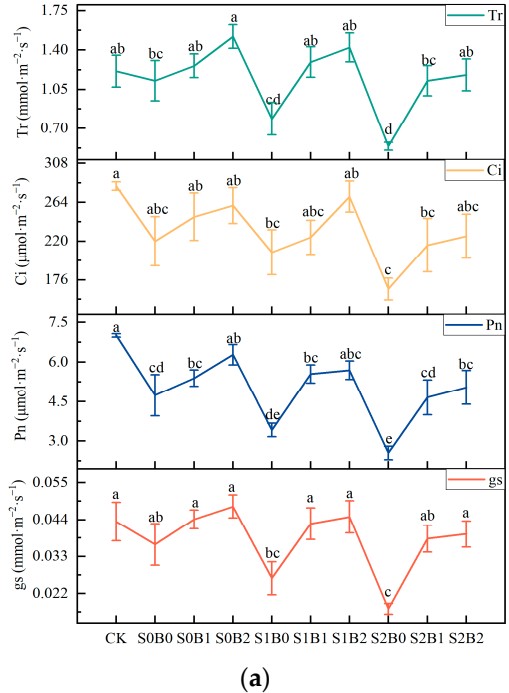
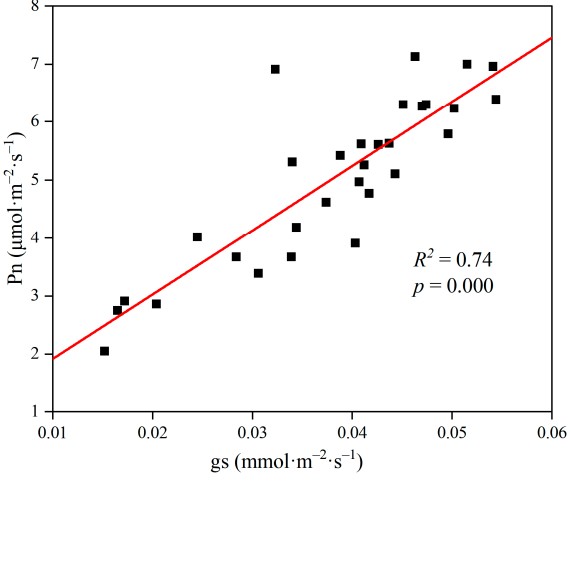

(**a**)　　　　　　　　　　　　　　　　　　　　　　　　(**b**)

**Figure 4.** Effect of salt stress and biochar application on (**a**) cabbage gas exchange parameters; (**b**) the relationship between stomatal conductance (gs) and net photosynthetic rate (Pn) was presented with a linear regression at $p < 0.001$ level. Bars followed by the same letter are not significantly different ($p < 0.05$). Error bars indicate standard error of the means (SE). S: salt concentration; B: biochar application; S × B: their interactive effect.

### 3.7. PCA and Multiple Regression Analysis

The mitigation effect of the biochar on the saline stress was evaluated by combining eight indicators in the principal component analysis. Four principal components (PC1s, PC2s, PC3s, and PC4s) were extracted from the saline-stress and no-biochar treatments, which together explained 91.01% of the data variability (Table 3). PC1s predominantly consisted of the ΦPSII, gs, and Pp. PC2s mainly consisted of the soil $Na^+$ and $K^+$ contents. PC3s comprised the cabbage $Na^+/K^+$ ratio, and PC4s consisted of the soil volumetric water content. The cumulative variance contribution of the four principal components (PC1$_B$, PC2$_B$, PC3$_B$, and PC4$_B$) in the biochar treatments was 83.82% (Table 3). PC1$_B$ mainly included the soil $Na^+$ and $K^+$ contents, PC2$_B$ included the Pp, PC3$_B$ included the cabbage $Na^+/K^+$ ratios, and PC4$_B$ included the ABA content.

**Table 3.** Result of principal component analysis.

| Principal Components | PC1 | | PC2 | | PC3 | | PC4 | |
|---|---|---|---|---|---|---|---|---|
| Biochar | N | Y | N | Y | N | Y | N | Y |
| Total eigenvalue | 3.034 | 2.412 | 1.733 | 1.888 | 1.458 | 1.398 | 1.063 | 1.009 |
| Percent (%) of variance | 37.927 | 30.145 | 21.662 | 23.594 | 18.221 | 17.471 | 13.282 | 12.612 |
| Cumulative (%) | 37.927 | 30.145 | 59.589 | 53.739 | 77.811 | 71.211 | 91.012 | 83.822 |
| Eigen vectors | | | | | | | | |
| ΦPSII | −0.862 | −0.448 | 0.278 | −0.561 | −0.256 | 0.406 | 0.030 | 0.341 |
| gs | −0.907 | 0.596 | 0.042 | 0.481 | −0.160 | −0.160 | 0.083 | 0.432 |
| Pp | 0.732 | 0.520 | −0.473 | −0.593 | 0.032 | −0.466 | 0.104 | −0.124 |

**Table 3.** *Cont.*

| Principal Components | PC1 | | PC2 | | PC3 | | PC4 | |
|---|---|---|---|---|---|---|---|---|
| ABA | 0.518 | 0.506 | −0.290 | 0.323 | −0.602 | 0.225 | −0.055 | 0.590 |
| $Na^+/K^+$ | −0.040 | −0.357 | −0.118 | 0.538 | 0.725 | 0.663 | −0.593 | −0.221 |
| $Na^+$ (s) | 0.527 | 0.618 | 0.791 | −0.452 | −0.269 | 0.545 | −0.04 | −0.228 |
| $K^+$ (s) | 0.476 | 0.782 | 0.840 | −0.238 | 0.242 | 0.442 | 0.006 | −0.069 |
| SWC | 0.045 | 0.457 | −0.027 | 0.578 | 0.479 | −0.086 | 0.829 | −0.487 |

N: no biochar treatment; Y: biochar treatment.

To further clarify the degree of the influence of the plant osmoregulation, water status, and hormone levels on the gs, a multiple linear regression analysis was performed to calculate the standardized partial correlation coefficients. The higher the coefficient between each index and the gs, the greater its influence on the gs. As shown in Figure 7, the partial correlation coefficients between the $Na^+/K^+$ ratio and gs were 0.702, 0.485, and 0.332.

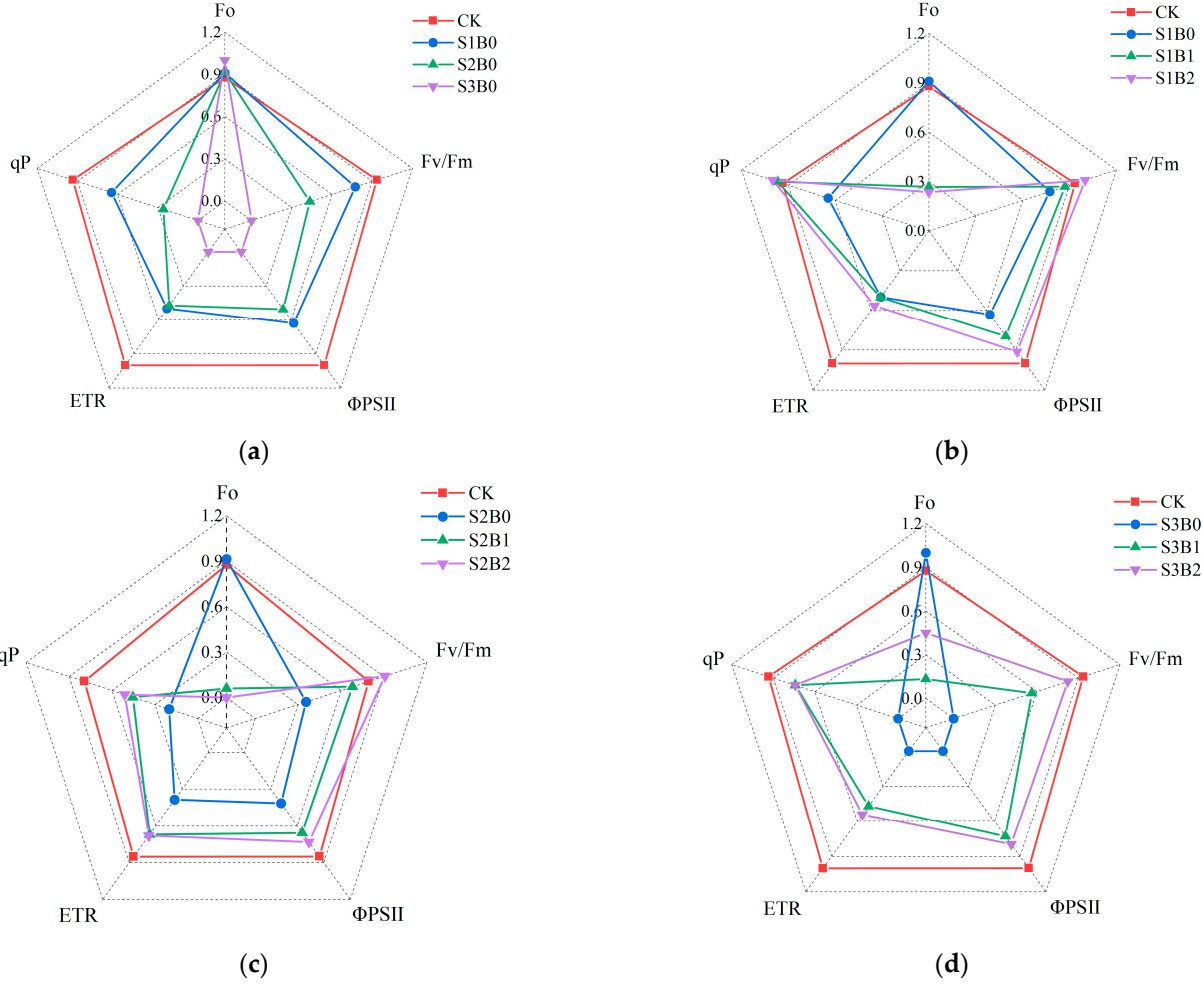

**Figure 5.** Chlorophyll fluorescence parameters (Fo, Fv/Fm, ΦPSII, ETR, and qP) for (**a**) salt stress without biochar treatments; (**b**) 25 mM salt concentration and biochar addition treatments; (**c**) 50 mM salt concentration and biochar addition treatments; and (**d**) 100 mM salt concentration and biochar addition treatments. All indicator data in the graphs are normalized to give a range of 0–1.

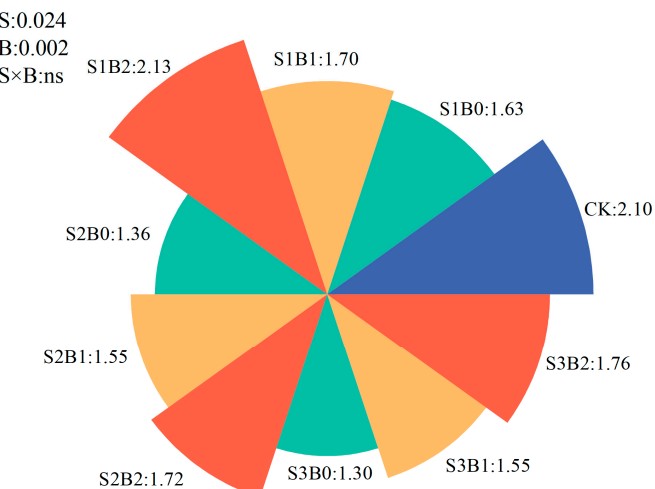

**Figure 6.** Effects of biochar and salt stress on yield (g/plant) of cabbage. The upper left digit represents the *p* value of the analysis of variance; ns indicates not significant, *p* > 0.05.

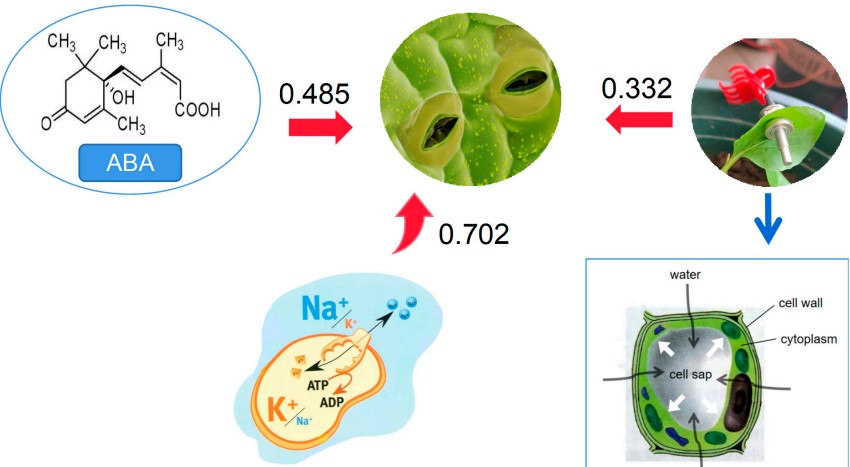

**Figure 7.** Degree of effect of ABA, $Na^+/K^+$ and Pp on gs. The numerical value in the figure represents the absolute value of the standardized partial correlation coefficient.

## 4. Discussion

Salinity is one of the main abiotic stresses that limits plant growth and development [34]. The photosynthetic performance of crops changes under salt-stress conditions. The identification of the physiological change mechanism of Chinese cabbage under salt stress is necessary to alleviate the stress damage and increase the carbon yield. In this study, we found that salt stress had an adverse effect on the fluorescence parameters of the Chinese cabbage, which resulted in the reductions in the ΦPSII and ETR, which demonstrated that the photosynthetic system of the Chinese cabbage was damaged under salt stress. The disruption of the photosynthetic apparatus is due to the low electron transport by PSII or to the structural failure of the light collection complex caused by the accumulation of reactive oxygen species (ROS) under salt stress [35]. However, the decrease in photosynthesis may be attributed to the loss of the stomatal function caused by salt, as we observed a strong correlation between the gs and Pn ($R^2 = 0.74$, *p* < 0.001). The point is confirmed by Bharath et al. [36], who reported that the stomatal closure of salt-stressed plants reduced the intercellular $CO_2$ concentration and limited the plant photosynthesis and nutrient absorption. This conclusion is supported by the correlation analysis in this study, in which the gs was positively correlated with the net photosynthetic rate. Therefore, the stomatal

opening is very important for the photosynthesis of Chinese cabbage. Stomatal closure directly leads to a decline in photosynthesis, which leads to a reduction in the plant yield.

Similar to nanofertilizers, biochar can release specific nutrients that are more efficient for plant growth [9]. The application of biochar to saline soils is believed to improve the stomatal opening in plants [37]. In the present study, the results of the PCA showed that gs and ΦPSII were not included in the indicative index after biochar was applied, which may be because the gs and photosynthetic capacity of the Chinese cabbage could be improved and maintained in a more stable state after the biochar addition. Similarly, Nahhas et al. [38] and Lashari et al. [39] also reported improvements in the gs and photosynthetic rates in faba bean and corn after biochar application to the saline soil. The positive changes in the photosynthetic capacity might be due to the positive effects of biochar on the chemical and hydraulic signals in Chinese cabbage, which increase the stomatal opening [40,41].

Stomatal movement is related to many factors [42,43]. Using the PCA method, we found that the change in the gs was mainly attributable to the effect of the biochar on the $Na^+$ and $K^+$ contents, water status, and ABA content (Table 3), which was mainly related to the biochar mitigation of the water stress and ion toxicity of the salt-stressed plants [44]. The addition of biochar can promote the plant roots' absorption of more water and reduce the ABA content, thereby maintaining the stomatal opening [37]. Furthermore, we found that the $Na^+/K^+$ ratio had the most pronounced effect on the gs, which was mainly related to the effects of the biochar on the contents of $Na^+$ and $K^+$ in the soil and plants. This finding differs from those of other similar studies, which have shown that ABA plays a pivotal role in the regulation of plant responses to salt stress [14]. The high osmotic potential caused by salt stress leads to an increase of the ABA contents in plants, which induces stomatal closure [45].

The biochar had a significant effect on the $Na^+$ content in saline soil; however, it had a weak effect on the $K^+$ content in soil. This result may be connected to the type of biochar. In a study on the effect of biochar on the soil K content, Fariba et al. [46] demonstrated that wheat biochar had a higher K concentration than corn biochar, and thus it had more of an impact on the availability of the soil K content. Further, the average response rate of soil-available potassium to biochar derived from crop residue was lower compared to biochar derived from other materials [47]. However, in this experiment, the $K^+$ content increased in the soil treated with biochar, which is because potassium occurs as an inorganic form in organic materials, and $K^+$ rapidly enters the soil solution after the biochar application to soil [48]. With respect to the $Na^+$ concentration in the soil, the data showed a difference between the application of 2% and 4% biochar, and possibly because biochar contains soluble salt. When the amount of biochar increases, the amount of soluble salt added to the soil also gradually increases. The increased amount of exchangeable Na and K in the soil could originate from the added biochar, which may reabsorb Na after 2 months, according to Nong et al. [49] and Nguyen et al. [50]. Some researchers believe that the plant response to soil quality improvement is predominantly reflected in the reduction in the $Na^+/K^+$ ratio in plant tissues [51]. Biochar ensures a low $Na^+/K^+$ content in plants by influencing the soil, which contributes to plant growth under salinized conditions and protects the plant metabolism by maintaining the cell turgor and membrane permeability [52].

Osmotic adjustment is an important mechanism that enables plants to tolerate salt stress [53]. Inorganic solutes have been shown to significantly contribute to osmotic regulation in a variety of plants [54]. $Na^+$ and $K^+$ are important inorganic osmoregulatory ions. Our results clearly showed that saline stress had pronounced adverse effects on the soil and cabbage. The $Na^+$ content of the cabbage significantly increased, whereas the $K^+$ content decreased, which resulted in a plant ion imbalance. A low $Na^+/K^+$ ratio in plants is an important physiological criterion for plant salt tolerance [55]. The reported data highlight that the application of biochar reduced the $Na^+/K^+$ ratio in the plants, thereby allowing the Chinese cabbage to undergo osmotic regulation and cell expansion [56]. This conclusion is supported by the view of Dourado et al. [57], who demonstrated that an increase in the osmotic adjustment is related to the balance of Na and Cl versus compatible

solutes. The positive effect of biochar on the $Na^+/K^+$ ratio in plants may be attributed to the following three mechanisms [58,59]: (1) the sodium absorption of biochar can retain sodium in the soil and reduce the plant absorption of $Na^+$; (2) the water retention of biochar can dilute the soil salt and decrease the concentration of $Na^+$; (3) biochar can increase the content of $K^+$ in soil, thus reducing the plant absorption of $Na^+$, which results in a lower $Na^+/K^+$ ratio.

In addition, the leaf hydraulic signal is another factor that affects the gs [60]. In this study, a leaf patch clamp pressure probe was introduced for the first time under conditions of salt stress and biochar, and the leaf turgor parameter (Pp) was used to reflect the water statuses of the plants. A significant increase in the Pp values was observed in the salt-treated plants, in contrast with the untreated control plants. The application of biochar led to a significant reduction in the Pp values, which indicated that the application of biochar can alleviate osmotic stress and promote the recovery of plant water [61]. Ache et al. [62] reported that the Pp is related to the stomatal movement. An increase in the Pp value indicates stomatal opening, whereas a decrease in the Pp value indicates stomatal closing. In this study, the effect of the Pp on the gs was weaker than that of the $Na^+/K^+$ (0.332), which may be because the regulatory effect of the plant hydraulic signals on the gs was relatively weak under the conditions of this experiment. The finding is explained by Turner et al. [16] (p. 13), [53] (p. 15), who demonstrated that the turgor pressure could only be partially maintained because the accumulation of solutes could not fully balance the water potential and the reduction in the cell turgor pressure. Therefore, the stomatal aperture (or gs) may be reduced with the incomplete recovery of the guard cell turgor pressure [63]. However, plants can maintain stomatal opening and other physiological activities by improving their osmotic regulation abilities.

Given that the soil water content in this experiment was measured after harvesting, it is inconsistent with previous conclusions that state that biochar can increase the effective soil water content [64]. According to the results of this study, there was no significant difference in the soil volumetric moisture content in any treatment, which could be because the irrigation with fresh water every two days in the experiment did not significantly affect the soil moisture, and it may be related to the soil texture. The meta-analysis of the effect of the biochar on the soil water retention showed that for the fine-textured soils, the field capacity remained unchanged [65]. In addition, the influence of external conditions on the soil quality should be evaluated by integrating various physical and chemical indexes [66]. The next step is to determine a variety of soil properties through field experiments to determine the effectiveness of biochar in saline soils.

## 5. Conclusions

The higher salt concentrations in the soils caused water stress in the cabbage plants and increased the ABA and $Na^+$ contents, which limited the stomatal opening and disrupted the photosynthetic system, thereby reducing the production. The application of biochar was highly effective in ameliorating the negative effects of saline stress. Biochar treatment can reduce the contents of $Na^+$ and ABA in Chinese cabbage, and it can improve the absorption of water and $K^+$. Furthermore, the biochar maintained the leaf stomatal opening by improving the osmotic regulation ($Na^+/K^+$) and chemical (ABA) and hydraulic (Pp) signals of the Chinese cabbage. Among them, osmotic regulation ($Na^+/K^+$ ratio) had the greatest effect on the stomatal conductance. Simultaneously, the fluorescence parameters and yield of the Chinese cabbage treated with biochar were improved, and the cabbage also demonstrated stomatal opening and enhanced photosynthesis. The results of this study are conducive to further understanding of the improvement mechanism of biochar on salt-stressed plants, and they provide a scientific basis and theoretical reference for the mitigation of soil salinization in facility vegetables. We suggest that biochar can be used in Chinese cabbage to alleviate the harm caused by soil salinization. Although biochar is effective in the remediation of saline soils and in improving the growth and physiological conditions of plants under a variety of environmental stresses, the potential impacts of

biochar on agricultural crops and their yields may vary due to the particular effects of biochar on the soil types. Therefore, more field studies are needed on different types of biochar to examine the effectiveness of biochar application on different soils.

**Author Contributions:** Conceptualization, L.Z.; methodology, R.C. and L.Z.; validation, R.C., L.Z., X.L. and J.Z.; formal analysis, R.C. and J.Z.; writing—original draft preparation, R.C.; writing—review and editing, J.M., X.L. and L.Z.; visualization, R.C. and L.Z.; supervision, R.C., J.Z. and L.Z.; project administration, L.Z. and J.M. All authors have read and agreed to the published version of the manuscript.

**Funding:** This research was funded by the National Natural Science Foundation of China (52079085, 52109061).

**Institutional Review Board Statement:** Not applicable.

**Informed Consent Statement:** Not applicable.

**Data Availability Statement:** Not applicable.

**Conflicts of Interest:** The authors declare no conflict of interest.

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
