# Peer review of "Biochar Application Maintains Photosynthesis of Cabbage by Regulating Stomatal Parameters in Salt-Stressed Soil"

_sustainability, doi:10.3390/su15054206_

Round 1

Reviewer 1 Report

In this manuscript, the authors evaluated the effects of biochar on salt stress through analyzed the physiological indexes of Chinese cabbage. The main advantage of this study is the systematic analysis of different factors that influence stomatal conductance. Moreover, the non-invasive leaf patch clamp pressure (LPCP) method was used in this study to continuously monitor the water status of plants.

At the same time, there are some limitations in this paper. There are mainly the following points.

1. Line 34: Should the citations for the sentence "Leaf stomata are important for water uptake and carbon assimilation in plants" be 5 and 6 ?

2. At present, there are many researches on leaf turgor. What are your innovations?

3. Why did the authors use that BC rate?

4. This paper did not introduce the effect of salt stress duration on the physiology of Chinese cabbage. Why did the author choose two photosynthetic data for analysis?

5. The effect of biochar application can be determined by applying the same PCA to all data. Why did the authors use two PCA analyses?

6. Line 288: The corresponding table in PCA should be Table 3

Line 364: References should be added to the argument of Nong et al.

Author Response

1. Line 34: Should the citations for the sentence "Leaf stomata are important for water uptake and carbon assimilation in plants" be 5 and 6 ?

ANSWER: Thank you for your advice. This issue has been revised in the article.

2. At present, there are many researches on leaf turgor. What are your innovations?

ANSWER: Thank you for your question. Firstly, there are many researches on leaf turgor pressure at present, but most of them focued on the spatio-temporal variation of turgor pressure, and few studies on the relationship between turgor pressure and stomatal conductance. Secondly, the innovation of this study is to observe the change of turgor pressure under the condition of salt stress and biochar application, and to analyze the relationship between turgor pressure and stomatal conductance.

3. Why did the authors use that BC rate?

ANSWER: Thank you for your question. Biochar application volume was determined by reviewing the literature. Leila et al. (2019) found that application of BC produced from mulberry wood, especially at 2% in the soil under salinity stress, reduced electrolyte leakage and increased the amounts of biomass. The biochar selected in this experiment was made from corn straw. Considering the different biochar types, it was wanted to determine whether the same and doubled application results were consistent. Therefore, 2% and 4% biochar application rates were selected.

Leila, M., Mohammad, Moghaddam., and Amir, L. (2019). Alleviating negative effects of salinity stress in summer savory (Satureja hortensis L.) by biochar application. Acta Physiologiae Plantarum 41:98

4. This paper did not introduce the effect of salt stress duration on the physiology of Chinese cabbage. Why did the author choose two photosynthetic data for analysis?

ANSWER: Thank you for your question. Firstly, previous studies have shown that the limiting factor of plant photosynthetic decline is related to the duration of salt stress. With the extension of salt stress duration, stomatal restriction would be changed to stomatal and biochemical restriction. Therefore, in order to illustrate the importance of stomatal opening to photosynthesis, two photosynthetic data were selected to analyze the limiting factors of photosynthetic decline. Secondly, according to the suggestions of reviewers and considering that the focus of this study is to analyze the influencing factors of the application of biochar in saline soil on stomatal conductance, this part of the paper has been revised.

5. The effect of biochar application can be determined by applying the same PCA to all data. Why did the authors use two PCA analyses?

ANSWRE: Thank you very much for your advice. Firstly, The role of principal component analysis is to explore the core factors of change and determine the degree of influence of factors. Principal component analysis was used in this study to explore the core factors of changes caused by biochar application. Secondly, at present study, PCA1 was the analysis of data from treatments with salt stress and no biochar, and PCA2 was the analysis of data from treatments with salt stress and biochar. The use of PCA2 could demonstrate the effect of biochar application, consistent with your proposal to use the same PCA for all data. The purpose of using PCA1 was to make a comparison, so as to clarify which representative indicators are affected by the application of biochar, and then analyze the internal reasons for the mitigation effect of biochar on salt stress.

6. Line 288: The corresponding table in PCA should be Table 3

Line 364: References should be added to the argument of Nong et al.

ANSWRE: Thank you for your advice. This issue has been revised in the article.

Reviewer 2 Report

M$M

L143-144.  Was the plant harvested at maturity for leaf ABA, Na and K determination?

L158-159.  Yield per plant………was the root also considered as yield?

Results

L280-281.   Figure 6, Please include your yield values.

Discussion

L330-331.   ………resulted in gs.  Resulted in what? Please check senstence.

L360.      ‘She pointed out that’.  They pointed out that

Conclusion

L401.       ‘proved the improvement of improved gs

Reviewer 3 Report

In general the paper is good and should be published after major revisions. Dear authors, you should be remains to establish a cause and effect relationship, that is, to relate the characteristics of the biochar with the observed effects. Some points are highlighted below.

Several abbreviations placed on paper were not given the first time they appear like: PSII, ABA (Abstract), ABA inline. Use the full form the first time the abbreviation appears.

Line 41 – citation is incomplete – correct for Leila et al. [9], the request must be throughout the manuscript - please correct it.

Line 46 - The best word is retain insted fix

Line 47 - rewrite the sentence, it's confusing

Line 128 - The detailed the method used to extract the Na and K

Line 182 - Correct Na+

In table 2, The statistical approach is incorrect, because the author compares the treatments as single treatments and not as a factorial  scheme design analysis.

In Figure 6 there was no value caption, in this figure several information was lost, the author must be the approach in this figure.

How can the author indicate that the effect of biochar is related to the cited mechanisms? This characterization of the biochar is poor, thus, a lot of information was lost, the authors must prove the potassium contents in the biochar, because its content can be correlated with the observed effects.

The paragraph in line 348 is too long, please rewrite it.

Although there is no effect on the availability of K in the soil, the K content in the biochar is relevant for some materials, as the K content in the soil solution has increased, so again the author must prove the K content in the biochar.

Reviewer 4 Report

The study is addressing “Biochar application maintains the photosynthesis of cabbage by regulating stomatal parameters in salt-stressed soil”. Chinese cabbage, was selected for the pot experiment in this study. Soil and plant Na+ and K+ concentrations, water status, and plant stomatal parameters were measured. The research is very limited in the aspects of the study and covers only a few points of the whole study. The literature study is very weak and outdated, and there is no proper linkage between different sections of the papers. The language of the paper also needs further improvements as it contains several mistakes. The headings also need changes as they should be based on obtained results instead of the equipment used. The English language used in the manuscript needs improvements, as there are some punctuation and grammatical mistakes throughout the manuscript. Sentences need more clarity and better construction. Some figures need more transparency; special focus is required in labelling the axis and titles. Overall, the paper needs major structural and literature revisions to meet the requirements of the journal. It is obvious the quality of the manuscript does not meet the standards of the Sustainability Journal, therefore needs major revisions.

Specific comments:

1.      The information is scattered throughout the manuscript, major structural changes are required, please do proper paragraphing throughout the manuscript. Please remove bullet points from the entire manuscript.

2.      Abstract:  The abstract only contains some parameters without any process conditions or key values from results, which is insufficient to delineate the whole picture of the contribution and possible application of this study. It is suggested to add some background with a few objectives, key values from the results, possible applications of this study and highlight the novelty of this work clearly (200-250 words).

3.      Revise keywords add more specific and novel keywords with broader meanings (5-7 words).

4.      The introduction is a lack of sufficient background information, which is unable to give the reader detailed background knowledge and possible wide application of this study. The introduction needs to be more emphasized on the research work with a detailed explanation of the whole process considering past, present and future scope. It needs to be more emphasized in the research work with a detailed explanation of the whole process. Research gaps should be highlighted more clearly and future applications of this study should be added. 

5.      Section 2. Materials and Methods: Authors have not mentioned what standard methods have been followed, please support this section with recent literature.

6.      The graphs throughout the manuscript are not consistent with blurry resolution. Revise these graphs with high-quality images and consistent fonts. The units stated in the graphs axis need to be double-checked.

7.      Results and discussion should be combined as “Results and Discussion” one heading.

8.      Have the authors performed sensitivity analysis on the data of these experiments? How these results are authentic? If not, then please performed a detailed sensitivity analysis to support the originality of the results.

9.      Results and discussion section: The obtained values in the results are just stated in the text without explaining them. Explain the reasons behind your trends/values and discuss them critically with literature.

10.   The discussion presented is very weak no strong comparison has been made with the literature to support the authenticity of the obtained results. Therefore, the authors are suggested to discuss their results with the following recent researches about novel agricultural systems, nano fertilizers, osmotic adjustment, soil properties and related pollution effects to make the background and discussion more strong, the following recent studies should be added: Journal of the Saudi Society of Agricultural Sciences, 2022;21(8):525-535. Soil Research,2021;60 (6): 485-496.  Environmental Chemistry Letters, 2022; 20: 2709–2726.

11.   Units: Use SI units; follow the correct format (e.g., use M for mol/L; d for days, h for hours, min for minutes, etc.) Be consistent! Use uppercase L for l (litre).

12.   Avoid an abundance of references do not cite more than 2 references in a single place. Correct all these types of references throughout the manuscript. 

13.   Comprehensively write the conclusions and should contain key values, suitability of the applied method, the major findings, contributions and possible future outcomes (300 words).

14.   Please revise the tables and figures captions for a better understanding. Should be comprehensive and meaningful.

15.   The reference style is not according to the Sustainability Journal, please correct the references styles. The authors must read the guide for authors.

16.   The authors are advised to revise references, including the latest references. Please see some suggestions in the specific comments and for the ‘introduction’ section.

17.   Some typo errors are present, the authors must check these carefully before submitting the manuscript. 

Round 2

Reviewer 1 Report

No.

Reviewer 3 Report

The paper should be accepted.

Reviewer 4 Report

The authors have addressed most of the comments; they have also tried to make changes according to the reviewers’ suggestions. After revisions, the quality of the manuscript has been adequately enhanced. Therefore, the manuscript could be considered for publication in the Journal.